# Polymer-Based Hydroxyapatite–Silver Composite Resin with Enhanced Antibacterial Activity for Dental Applications

**DOI:** 10.3390/polym16142017

**Published:** 2024-07-15

**Authors:** Jesús Alberto Garibay-Alvarado, Diana Juana Garcia-Zamarron, Pamela Nair Silva-Holguín, Alejandro Donohue-Cornejo, Juan Carlos Cuevas-González, León Francisco Espinosa-Cristóbal, Álvaro de Jesús Ruíz-Baltazar, Simón Yobanny Reyes-López

**Affiliations:** 1Laboratorio de Materiales Híbridos Nanoestructurados, Departamento de Ciencias Químico-Biológicas, Instituto de Ciencias Biomédicas, Universidad Autónoma de Ciudad Juárez, Envolvente del PRONAF y Estocolmo s/n, Ciudad Juárez 32300, Mexico; jesus.garibay@uacj.mx (J.A.G.-A.); al231304@alumnos.uacj.mx (P.N.S.-H.); aruizbaltazar@fata.unam.mx (Á.d.J.R.-B.); 2Programa de Maestría en Ciencias Odontológicas, Departamento de Estomatología, Instituto de Ciencias Biomédicas, Universidad Autónoma de Ciudad Juárez (UACJ), Envolvente del PRONAF and Estocolmo s/n, Ciudad Juárez 32310, Mexico; diana.zamarron@uacj.mx (D.J.G.-Z.); adonohue@uacj.mx (A.D.-C.); juan.cuevas@uacj.mx (J.C.C.-G.); leon.espinosa@uacj.mx (L.F.E.-C.); 3CONAHCYT-Centro de Física Aplicada y Tecnología Avanzada, Universidad Nacional Autónoma de México, Boulevard Juriquilla 3001, Santiago de Querétaro 76230, Mexico

**Keywords:** resin, hydroxyapatite, silver, nanoparticles, antibacterial

## Abstract

The primary objective of this investigation was to synthesize a resin incorporating nanoparticles of hydroxyapatite and silver (HA-NpsAg) to enhance biocompatibility and antimicrobial efficacy, thereby facilitating potential implementation within the dental industry. These enhancements aim to ensure reliable, durable, functional, and aesthetically pleasing restorations while concurrently reducing susceptibility to bacterial colonization within the oral cavity. Hydroxyapatite powders were prepared using the sol–gel method and doped with silver nanoparticles obtained by chemical reduction. The crystalline amorphous calcium phosphate powder had a particle size of 279 nm, and the silver nanoparticles had an average diameter of 26.5 nm. Resin spheres containing HA-NpsAg (RHN) were then synthesized at two concentrations (0.5% and 1%) by dissolving the initial monomer mixture in tetrahydrofuran. Subsequent antimicrobial evaluations were conducted via agar diffusion and turbidimetry, employing three strains of Gram-negative bacteria (*E. coli*, *K. oxytoca*, *and P. aeruginosa*) and three strains of Gram-positive bacteria (*S. mutans*, *S. aureus*, and *B. subtilis*). The findings revealed that *P. aeruginosa* exhibited maximum susceptibility to RHN powder at a concentration of 0.5%, while RHN powder at 1% concentration demonstrated maximal inhibition against *S. aureus* and *S. mutans.* Overall, our study highlights the successful synthesis of a dental resin with hydroxyapatite and silver nanoparticles, exhibiting bactericidal properties at low silver concentrations. These findings hold promise for enhancing dental materials with improved antimicrobial efficacy and clinical performance.

## 1. Introduction

Oral bacterial infections are a recurrent issue since the mouth has ideal characteristics for bacterial development. Caries is one of the most common infectious diseases worldwide [1], and it could be easily treated through restorative dentistry [2]. Composite resins are the most used materials for reconstruction of damaged teeth, and while aesthetic and mechanical characteristics are well established, most of them do not possess antibacterial activity, which could be very helpful in preventing secondary caries and restoration failure [3]. Different approaches have been taken to produce a material capable of being used in restorative dentistry with characteristics such as antibacterial activity and the capacity of regeneration. Some of the alternatives include the use of quaternary ammonium monomers [4], chlorhexidine [5], zinc oxide [6], and silver nanoparticles [7].

Great surface area to volume ratio provides metal nanoparticles the ability to inflict antibacterial action with small concentrations [8]. Various studies have shown that there is a correspondence between the size of the nanoparticle and the antibacterial activity, as smaller particles present higher inhibition [9,10,11]. Nanoparticle antibacterial action has not been completely understood, but multiple mechanisms may be responsible and could be working simultaneously. The disruption of ion channels, damage in the bacterial membrane and cytoplasm leakage, oxidative stress, and protein architecture disruption are some of the mechanisms involved in bacterial death [12]. Silver nanoparticles have a broad spectrum and bacteria do not develop resistance, a useful quality in today’s medical environment as bacterial resistance to antibiotics is rampant.

Recent advancements in nanotechnology have spotlighted silver nanoparticles (AgNPs) for their dual role in medical applications. Notably, polyvinylpyrrolidone-stabilized AgNPs have shown efficacy as a suture coating, enhancing both antibacterial properties and suture durability [13]. Additionally, innovative bioinspired dressings combining AgNPs with chitosan, collagen, and genipin in a hydrogel matrix have emerged, promoting wound healing and tissue regeneration [14]. These advancements underscore the versatility of AgNPs in various biomedical applications, highlighting their role not only in direct antibacterial action but also in supporting tissue repair and recovery. The integration of AgNPs into complex matrices and medical devices represents a forward leap in addressing the challenges posed by antibiotic-resistant bacteria in clinical settings [15].

Hydroxyapatite is a well-known bioactive ceramic capable of improving tissue regeneration, with excellent compatibility. It is widely used in orthopedics and dentistry, and in the latter, applications such as filler where bone is lost, and the capacity to remineralize the teeth enamel have been extensively studied [16]. Studies carried out on the capacity of hydroxyapatite showed that it can fill up pores on teeth enamel and serve as a template for further mineralization [17], that it can remineralize enamel and cementum in the margins of a restoration [18], and that it can mineralize enamel in the same capacity or better as fluoride, making it an alternative [19,20]. Some studies have used micro- and nanohydroxyapatite in restorative glass ionomer-modified resin in the hopes of analyzing microleakage. Silver and hydroxyapatite used in conjunction are a promising choice in restorative dentistry, as they have been added to commercial orthodontic adhesives to improve the antibacterial and mineralization properties and, thus, the post-bracket-removal integrity and aesthetics of teeth [21,22,23]. Attempts at incorporating Ag and HA at restorative resin have also been made; however, the process involved the creation of a fully functional ceramic-less Bis-GMA/TEGDMA-based resin with great antibacterial performance [24]. The incorporation of Ag, HA, and even fiberglass has also been attempted [25,26]. Silver is widely known for its antibacterial activity; however, its incorporation into a composite resin requires that the amount is both small enough that it does not change its aesthetic properties or has a greatly negative impact on human cells, and sufficiently large that it can maintain antibacterial, or, at the very least, bacteriostatic activity [27,28,29].

Bacterial resistance affects the effectiveness of treatments and increases the burden of disease. It is essential to implement strategies to prevent and control resistance, optimizing the use of compounds with antimicrobial agents in the prevention of infections. *Staphylococcus aureus* is a bacterium that causes a wide variety of infections, from skin infections to serious infections such as pneumonia and sepsis; its resistance to antibiotics, especially methicillin, has led to the development of resistant strains (MRSA), which are difficult to treat. *Bacillus subtilis* is a bacterium widely used in industry and research. Although it is not pathogenic for humans, its resistance could affect the production of enzymes and other biotechnological products. *Escherichia coli* is common in the human intestine, but some strains can cause serious infections; their resistance to antibiotics makes it difficult to treat urinary tract infections, gastroenteritis, and sepsis. *Streptococcus mutans* is the main cause of dental caries, and its resistance to antibiotics affects the treatment of oral infections. Pseudomonas aeruginosa is common in nosocomial infections, especially in immunocompromised patients. Resistance to multiple drugs makes its treatment difficult. *Klebsiella oxytoca* is the cause of urinary tract infections, pneumonia, and sepsis; hence, its resistance to antibiotics is worrying [30]. The objective of this study was to modify a commercial dental resin in which nanoparticles of hydroxyapatite and silver were incorporated to provide antimicrobial properties against Gram-positive and Gram-negative bacteria.

## 2. Materials and Methods

Hydroxyapatite synthesis

The hydroxyapatite was synthesized using a modified version of the method used by Garibay et al. (2021) [26]. Calcium nitrate tetrahydrate (99.0%, Sigma-Aldrich, Saint Louis, MO, USA) was dissolved in ethanol, and the triethyl phosphite (98.0%, Sigma-Aldrich, Saint Louis, MO, USA) was hydrolyzed in ethanol. The calcium nitrate solution was added dropwise to the triethyl phosphite solution and mixed for 1 h. The solution was vigorously stirred for 24 h at 40 °C, then aged for 6 h at 60 °C. The resulting sol was heated for 24 h at 100 °C to desiccate in an oven (Heratherm OGS60, Thermo Scientific, Lengensenbold, Germany), then calcined at 950 °C for 3 h using a ramp of 5 °C/min in a furnace (Thermolyne FB1410M, Thermo Scientific, Asheville, NC, USA) The solid was pulverized using an agate pestle and mortar.

Silver nanoparticle synthesis

The method of choice was chemical reduction. AgNO_3_ (99.0%, Sigma-Aldrich, Saint Louis, MO, USA) was used as metallic precursor, dissolved in deionized water to a concentration of 10 mM. Gallic acid (98.0%, Sigma-Aldrich, Saint Louis, MO, USA), the reducing agent, was then mixed with the silver nitrate solution under vigorous stirring. The pH of the resulting solution was adjusted to 11 units using a 1 M sodium hydroxide solution (97% Fisher Scientific, Fair Lawn, NJ, USA) [27].

Hydroxyapatite–silver nanoparticle doping

The hydroxyapatite–silver particles were produced by adsorption. Every 1 g of the hydroxyapatite powder was mixed in 100 mL of silver nanoparticles solution to obtain a concentration of 50 mM of silver. The mixture was stirred for 96 h at 25 °C, then vacuum filtered. The resulting product was dried for 24 h at 100 °C followed by 3 h at 200 °C [28].

Hydroxyapatite–silver-composite resin

The composite resin (3M ESPE Filtek Z350 XT A3B, 3M, St. Paul, MN, USA) was diluted to 80% with tetrahydrofuran (99.9%, Sigma-Aldrich, Saint Louis, MO, USA) inside an amber glass vial, then mixed until homogeneous with the hydroxyapatite powder to obtain concentrations of silver of 0.50 mM and 0.25 mM. Discs of 5 mm in diameter and 2 mm in height were formed using both types of HA-AgNPs resin and plain resin, then photopolymerized until hard using a dental LED curing lamp with an intensity of 1500 mw/cm^2^ and a wavelength of 420 nm (AZDENT, University Science Park, Zhengzhou, China) placed at 1 cm from the surface of the sample during 10 s.

Characterization

The UV–Vis spectra were carried out at 25 °C using a UV-Vis spectrophotometer (Cary 100 UV–Vis, Varian, Mulgrave, Victoria, Australia) and a 10 mm quartz cell for both the hydroxyapatite powder and the silver nanoparticles. The silver and hydroxyapatite particle sizes were estimated by dynamic light scattering using a nanoparticle analyzer (Nanopartica SZ-100 analyzer, Horiba Scientific, Shanghai, China), using 20 mg of hydroxyapatite dissolved in 4 mL of water, while the silver nanoparticles were measured dissolving 5 μL of nanoparticle solution in 4 mL of water. Morphology of the hydroxyapatite and AgNPs was analyzed using a field emission scanning electron microscope (SU5000, Hitachi, Ichige, Japan). Infrared and Raman spectra were carried out using a ATR-FTIR spectrometer (Alpha Platinum, Bruker, Billerica, MA, USA) and a Raman spectrometer with 532 nm laser (WITec Alpha 300, Oxford Instruments, Abingdon, UK), scanning an area of 5 µm for the AgNPs, hydroxyapatite, and the resin alone and as a composite. X-ray diffraction patterns were obtained using a diffractometer (X’Pert PRO, Malvern Panalytical, Almelo, The Netherlands) for the hydroxyapatite powders alone and mixed with silver. The diffraction patterns were refined, and the chemical composition was calculated using the Profex software (Profex version 5.3.0, Nicola Döbelin, Solothurn, Switzerland) [29].

The measurement of the antibacterial activity of the HA-AgNPs composite resins was carried out using cultures of *Staphylococcus aureus* (ATCC 23235), *Bacillus subtilis* (ATCC 23857), *Escherichia coli* (ATCC 25922), *Streptococcus mutans* (ATCC 35668), *Pseudomonas aeruginosa* (ATCC 10145), and *Klebsiella oxytoca* (ATCC 49131). Three of the bacteria are Gram+ and three are Gram-, from which *S. aureus* and *E. coli* are universally used as models of each group, respectively. *S. mutans* is widely known for its role in dental caries, and *P. aeruginosa* and *K. oxytoca* are known to develop opportunistic infectious process in immunosuppressed patients [30]. A disk diffusion test was first performed as a qualitative measure. The bacteria were cultured in nutritive broth (BD Bioxon) for 18 h at 37 °C before the test. Then, 100 µL of standardized suspensions with an optical density of 0.1 according to the McFarland scale were inoculated into Müller-Hinton agar (BD Bioxon) plates. Three discs corresponding to each type of composite resin were embedded into the agar equidistantly. The antibacterial effect was determined by measuring with a caliper the clear zones around the composite discs, which would indicate the absence of bacterial growth. All tests for each microorganism were conducted in triplicate, the inhibition halo was averaged, and the standard deviation was calculated.

The minimum inhibitory concentration was determined through the broth microdilution method using a microplate reader (Multiskan MCC, Thermo Fisher Scientific, Shanghai, China). Using the same bacterial cultures as before, 200 µL of standardized inoculum of each bacteria strain were added into a 96-well microplate with 5 mg of each treatment. The microplate was incubated for 18 h at 37 °C under constant agitation, then the optical density was measured at 570 nm from time 0 and each half-hour afterwards until 8 h had passed. The measurements were carried out in triplicate. The percentage of inhibition was calculated for each time and, finally, all data were analyzed using the software SPSS Statistics (SPSS version 25, IBM, Armonk, NY, USA) using ANOVA and Tukey’s multiple comparison test, expressing results as “mean value ± standard deviation”, where a *p* value ≤ 0.05 was considered as statically significant.

## 3. Results

Characterization by IR and Raman spectroscopy was carried out to prove hydroxyapatite composition. In the IR spectra (Figure 1a) bands located at 560, 600, and 628 cm^−1^ are attributed to flexing vibrations of the PO_4_^3−^ group [31]. Three prominent bands at 961, 1022, and 1090 cm^−1^ correspond to a symmetric vibration of the PO_4_^3−^ group. The Raman spectra (Figure 1b) showed bands at 437, 587, 961, 1046, and 1079 cm^−1^. The band at 961 cm^−1^ is the most intense and is related to the PO_4_^3−^ (*v*1) vibration, typically associated with carbonated apatite and assigned to the presence of acid phosphate coming from the tricalcium phosphate. The bands located at 437 and 587 cm^−1^ are attributed to the flexural modes of the phosphate group, while the 1046 and 1079 cm^−1^ bands are related to the vibration modes of stretching of the phosphate group. At 1090 cm^−1^, it is related to a degenerated asymmetric stretching (*v*3) vibrational mode of the phosphate group and bending mode (*v*4) for phosphate, which is generally found at 578 cm^−1^ [27,31]. The XRD diffractogram (Figure 1c) was compared to the XRD pattern for pure hydroxyapatite (JCPDS 09-0432). Prominent peaks are shown at approximately 26, 32, 33, 34, 40, 46, and 50 2θ°, corresponding to the planes (002), (211), (300), (202), (310), (222), and (213), respectively, and the ratio of Ca/P was estimated in 1.7. The UV–Vis spectra of the hydroxyapatite in Figure 1d shows absorbance at 300 nm.

The micrography in Figure 1e shows a conglomerate of particles; in this example, the size of the particle protruding is 231.87 ± 51 nm. The elemental disposition on the material was made available through EDS, having mostly calcium, oxygen, and phosphorous. Some traces of carbon are visible, revealing the possible formation of carbonates, not uncommon during the synthesis of hydroxyapatite. The DLS analysis (Figure 1f) showed that the average diameter of the hydroxyapatite particles is 219 ± 24.2 nm. The STEM analysis confirmed the size of the particles but also revealed the morphology as spherical.

Silver nanoparticles

The UV–Vis spectra of the silver nanoparticles (Figure 2a) show strong absorbance at 407 nm, which coincides with the surface plasmon resonance of silver nanoparticles of small size [32]. At 286 nm, another band indicates the presence of the reducing agent (gallic acid) [33]. The size of the silver nanoparticles was measured through DLS, and the average diameter was 26.5 ± 5 nm (Figure 2b), with a very narrow distribution, well under 100 nm. The nanoparticles were in an aqueous solution (Figure 2c), yellow in color, which has been described in other studies for solutions with nanoparticles under 10 nm in size [34,35]. The nanoparticles were observed through TEM, showing they were not agglomerated, and that the morphology was spherical (Figure 2d). Various polysaccharides can be used as reducing, surface-modifying, and stabilizing agents in the synthesis of silver nanoparticles. Gallic acid has proven to be a good reducing and stabilizing agent, in addition to promoting a nontoxic green synthesis in the production of silver nanoparticles [28].

Hydroxyapatite with silver nanoparticles

The UV–Vis spectra of the hydroxyapatite–silver powder (Figure 3) was compared against the spectra of only hydroxyapatite (Figure 1a). At 294 nm, the strong peak corresponds to hydroxyapatite [31], whilst the small bump at 428 nm is related to the surface plasmon resonance of the silver nanoparticles [36]. The IR spectra of both HA powders with and without silver (Figure 4) show a band at 562 cm^−1^ corresponding to a flexing vibration of the PO_4_^3−^ [37]. A strong band at 1027 cm^−1^ corresponds to a symmetric vibration of the PO_4_^3−^. At 1655 cm^−1^ a small shoulder belonging to the OH^−^ group, most likely because of the presence of free water [26]. Two weak bands at 1427 and 873 cm^−1^ correspond to the CO_3_^2−^ group [38]. Independently of the presence or concentration of silver nanoparticles, there is no significant change in the spectra or the bands themselves, except for the fact that some became more defined.

Hydroxyapatite–silver nanoparticles composite resin

The IR spectrum of the cured composite resin discs with and without silver is shown in Figure 5. The spectrum presents a similar shape in the three samples: the broad band at 1110 cm^−1^ corresponds to siloxane vibrations [39], the band located in 1709 cm^−1^ is attributed to a carbonyl stretching on the resin, and, finally, a band is observed at the 450 cm^−1^ level, which corresponds to (CO_3_^2−^). For tetrahydrofuran, most intense bands appear at 749, 2861, and 1091 cm^−1^ [40]; however, on the resin spectra, these bands are not apparent, since a small amount of heat was produced during curing of the resin. In addition, the discs were not used immediately after curing; the solvent most likely evaporated, and, indeed, it was evaporated, since the viscosity of the resin needed to be reduced to incorporate the HA but was restored to shape the discs. Regarding the toxicity of the solvent, according to the ACS [41], THF is relatively nontoxic, and, while it was selected among different solvents to dissolve the resin, THF worked the best, so a different solvent, less toxic, could be used. The X-ray diffraction analysis of the composite resin samples (Figure 6) showed a strong predominance of peaks related to the planes (111), (200), (220), and (311) of zirconia (monoclinic) [42], and a wide amorphous bump between 20 and 27 2θ° possibly related to silica [43], since both ceramics are components of the commercially available composite resin [44]. No peaks for hydroxyapatite are observable on the diffractogram, most likely because of the concentration being very small.

To determine the inhibition capacity of the composite resins, a disc diffusion qualitative test was performed. The composite resin with HA did not have any inhibition on the bacterial growth. The composite with concentration of 0.25 mM Ag did not inhibit the growth of any of the bacterial cultures, except for *S*. *mutan*s, where the halo had an approximate diameter of 5.35 mm. A concentration of 0.50 mM Ag only inhibited the growth of *S*. *mutans*, *S*. *aureus,* and *K*. *oxytoca* (Table 1), but for the rest of the cultures, no inhibition was observed.

Figure 7 shows the behavior of the bacterial culture in the presence of the composite resin. In general, the tested Gram-positive and -negative bacteria had a negative response against the silver content. Both concentrations, 0.50 and 0.25 mM, of silver are effective at bacterial inhibition; however, the higher the content of silver, the better the inhibition, except in the case of *K*. *oxytoca*, where the concentration of 0.25 mM was most effective. The inhibition behavior of the HA-AgNPs composite resin in both concentrations is always statistically different for all bacterial cultures used.

It has been shown that silver in cationic form has limited utility as an antimicrobial agent due to the formation of precipitates with anionic species, due to the low solubility of its salts, which is why it is necessary to achieve an antimicrobial composition or material with a controlled continuous release mechanism of silver ions. Low concentrations are used in this research to demonstrate the competitiveness of the addition of hydroxyapatite and silver. Silver turns out to be the therapeutic answer to the emergence of antibiotic-resistant bacteria. The limitation of the use of silver is the toxicity caused by the high concentration of silver ions administered; therefore, when using metallic nanoparticles, the concentration of ions is reduced with a greater bactericidal effect.

## 4. Discussion

The method used to obtain hydroxyapatite is well established. The sol–gel method allows the obtainment of very pure, highly refined crystalline amorphous calcium phosphate powder; however, since calcination is needed to form hydroxyapatite, this in turn provokes the formation of particles of higher size (279 nm). Regarding the silver nanoparticles, an ample variety of papers is dedicated to the study of silver nanoparticles, coinciding in the nature of the surface plasmon resonance of such metal, which is located between 350 and 500 nm [45]. The wavelength at which the silver interacts with light, and the size of the nanoparticles have a direct impact on the color of the solution to the point that even colorimetric scales have been fashioned after this behavior [46]. For particles in the size of 10 nm or less, the solution that contains them becomes yellow. The DLS analysis shows that the size of the particles is distributed in a very narrow range, with an average diameter of 26.5 nm. Some studies [47] mention that hydroxyapatite has a high adsorption capacity and can capture metal ions with great force. This characteristic could be either positive, releasing the silver ions in a controlled manner, or negative, sequestering the silver. Once the hydroxyapatite particles adsorbed the silver nanoparticles, its color changed from white to gray; however, when mixed with the composite resin, the color of the resin was not visibly altered. This may be because of the small amount used, as larger concentrations of silver would have changed the composite to a darker shade of gray, and this may be an unwanted quality on a restoration resin.

The analysis of the composite resin with and without HA-AgNPs shows the presence of silicon, most likely in the form of silica and monoclinic zirconia, as it is widely known to be an ingredient in commercial composite resins to improve its mechanical resistance and aesthetic appearance [47]. After the disc diffusion test, it was apparent that the silver concentration of 0.25 mM was not enough to have antibacterial effect and that only *S. mutans* was susceptible to such a concentration. However, the 0.50 mM concentration was, indeed, effective against other bacteria. The fact that silver may be adsorbed on the hydroxyapatite [31,48,49,50] and incapsulated in the resin could prevent the Ag^+^ ions from being dispersed in the culture medium. The turbidimetric assay showed that the resin had antibacterial effect because of silver being released in the medium. This resulted in different degrees of antibacterial effectiveness, which is also influenced by the specific resistance of each bacterial strain. All the bacteria had a negative response to the resin in the medium, as any hydroxyapatite exposed in the resin could start dissolving and, in turn, release the Ag^+^ ions. The action of the silver nanoparticles on the bacteria has not been fully described; however, one of the mechanisms is believed to be responsible for modifying the structure of the cell membrane. This characteristic could be directly linked with the nanoparticle size as the smallest sizes have a higher chance of penetrating the membrane of the cells; thus, lower concentrations are required [48,49,50,51].

Table 2 shows a comparison between the results of this study and data obtained from other studies about the antibacterial effect of silver nanoparticles and silver–hydroxyapatite composites. As reported, the antibacterial activities gradually increased with increasing Ag content. Ag nanoparticles lead to an increase in the formation of reactive oxygen species (ROS) that leads to the destruction of the bacterial cells. The higher concentration of ROS has many effects on the bacteria, such as lipid peroxidation and lipid peroxidation affecting the bacterial membrane integrity. Some of the results in Table 2 show that when hydroxyapatite–silver is mixed into a polymeric matrix [52], the amount needed for an antibacterial effect is significantly larger than when hydroxyapatite is on a ceramic matrix; this is due to the encapsulation of the AgNPs and the hinderance in the liberation of Ag ions.

Advances in nanotechnology are opening new possibilities for treating diseases, especially for drug-resistant infections. Silver nanoparticles have demonstrated their antimicrobial properties and applications. Therefore, research continues in the search for new treatments and composites with silver nanoparticles that largely help to combat bacterial infections.

## 5. Conclusions

A dental resin with hydroxyapatite and silver nanoparticles, with bactericidal properties, was produced. The hydroxyapatite nanoparticles were obtained by the sol–gel method, and its composition coincides with an HA of high purity according to infrared, UV–Vis, Raman spectroscopy, and EDS. XRD results showed high crystallinity and a Ca/P ratio of 1.7 according to the Rietveld refinement. According to the morphological analysis by SEM, the size of the HA particles was approximately 250 nm. The silver nanoparticles obtained by chemical reduction had a size of 26.5 ± 5 nm and a spherical morphology, according to UV–Vis spectrophotometry, with surface resonance plasmon at 407 nm. Two kinds of resin samples were obtained and doped with hydroxyapatite and silver nanoparticles (RHN) at 0.5% and 1% from a concentration of 50 mM Ag. The bacterium *Streptococcus mutans* showed susceptibility to the RHN sample with 0.5%. The 1% RHN resin had greater antibacterial activity against *Staphylococcus aureus* and *Streptococcus mutans* bacteria. The composites obtained had an antibacterial effect when the concentration of silver was 0.5 mM, showing that a small amount of silver can be useful as an antibacterial aid, since the amount of metal that can be absorbed or ingested is always called into question due to bioaccumulation concerns. However, silver has a low toxicity in humans, especially such an amount, which is also not released at once, but slowly.

## Figures and Tables

**Figure 1 polymers-16-02017-f001:**
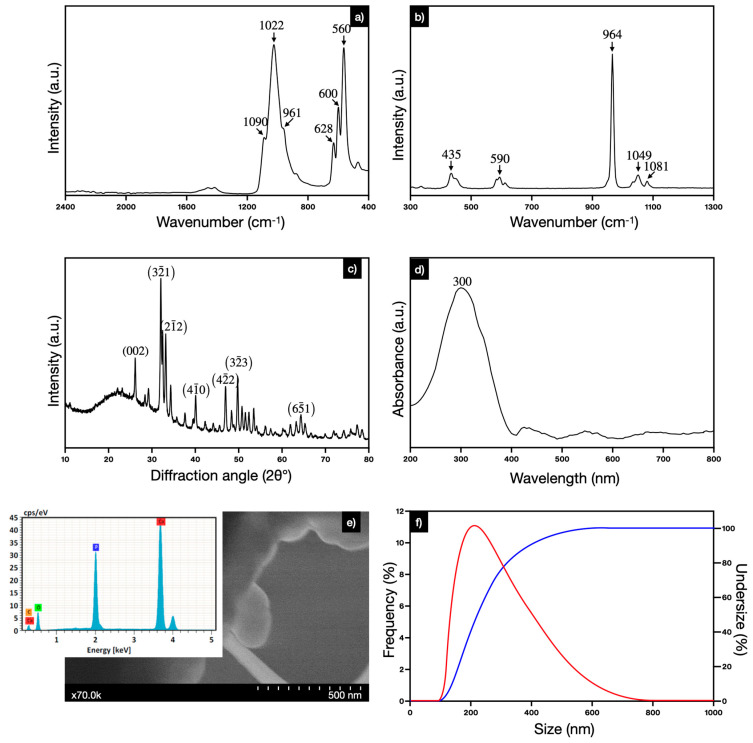
Characterization of HA powder: (**a**) IR spectra, (**b**) Raman spectra, (**c**) X-ray diffractogram, (**d**) UV–Vis spectra, (**e**) STEM micrography with EDX spectra, and (**f**) DLS size histogram of hydroxyapatite powder.

**Figure 2 polymers-16-02017-f002:**
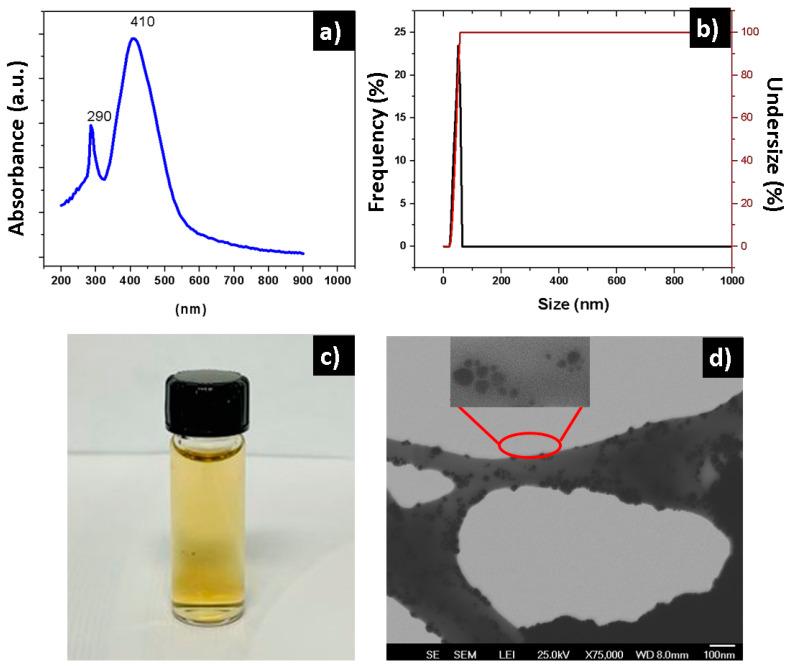
(**a**) Absorption UV–Vis spectra of the silver nanoparticles, (**b**) silver nanoparticles size histogram, (**c**) photograph of the nanoparticle solution, (**d**) and STEM micrography of the nanoparticles.

**Figure 3 polymers-16-02017-f003:**
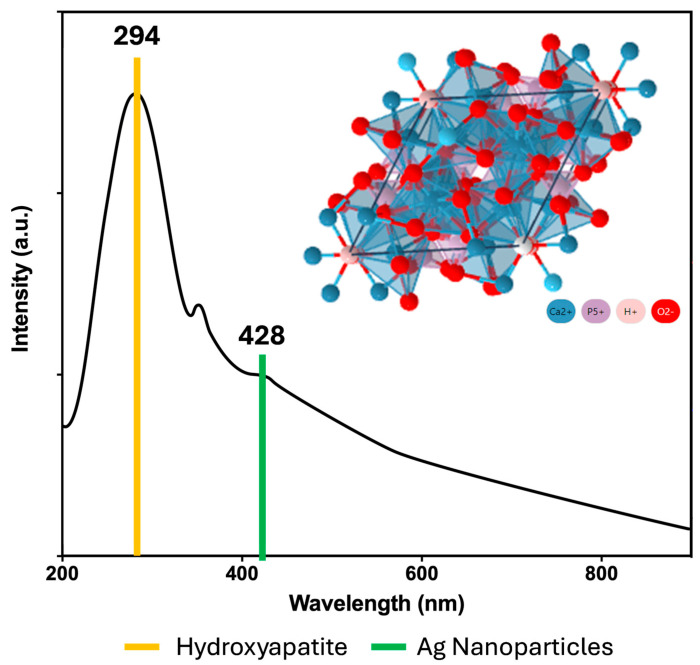
UV–Vis spectra of the hydroxyapatite–silver nanoparticles powder at 50 mM.

**Figure 4 polymers-16-02017-f004:**
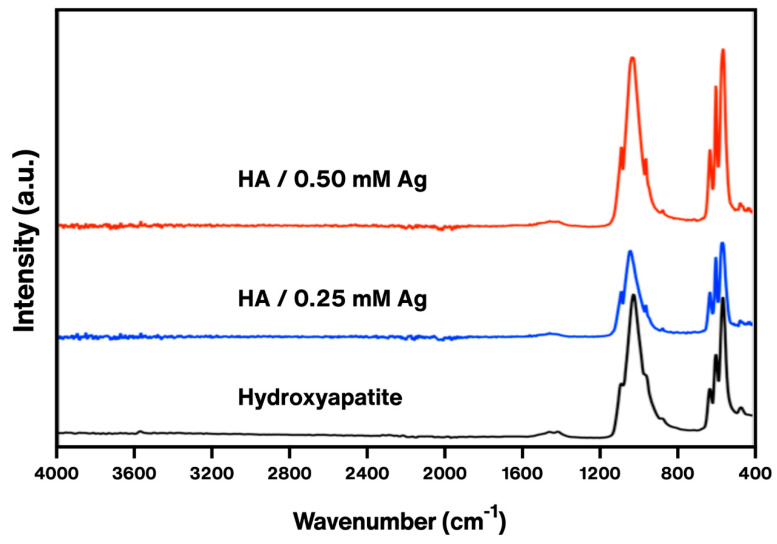
ATR-FTIR spectra of the hydroxyapatite and hydroxyapatite–silver nanoparticles powder (0.25 and 0.50 mM).

**Figure 5 polymers-16-02017-f005:**
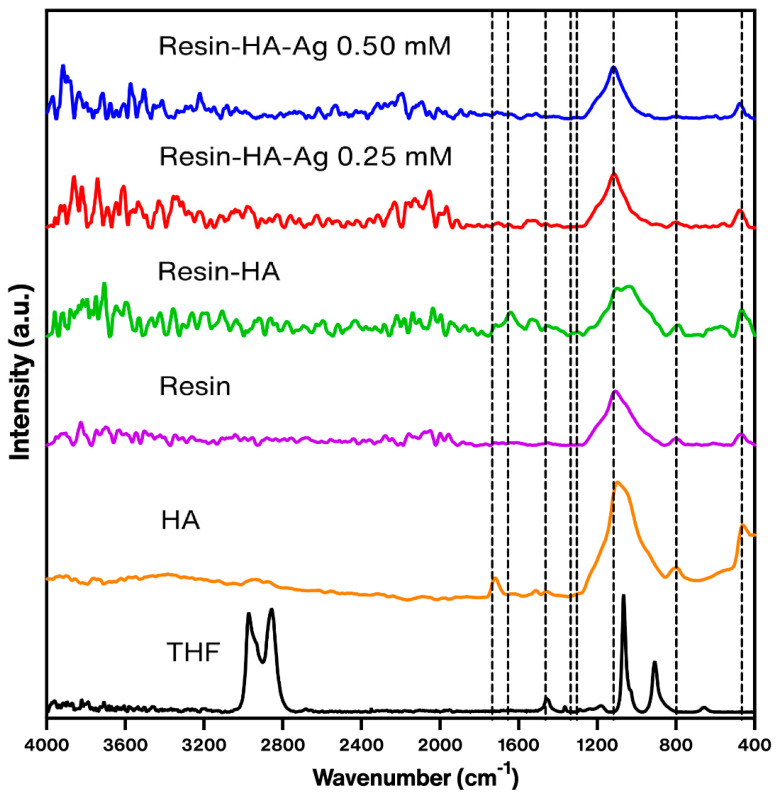
IR spectra of composite resin with 0.50 mM of silver (blue), 0.25 mM of silver (red), and with hydroxyapatite (black).

**Figure 6 polymers-16-02017-f006:**
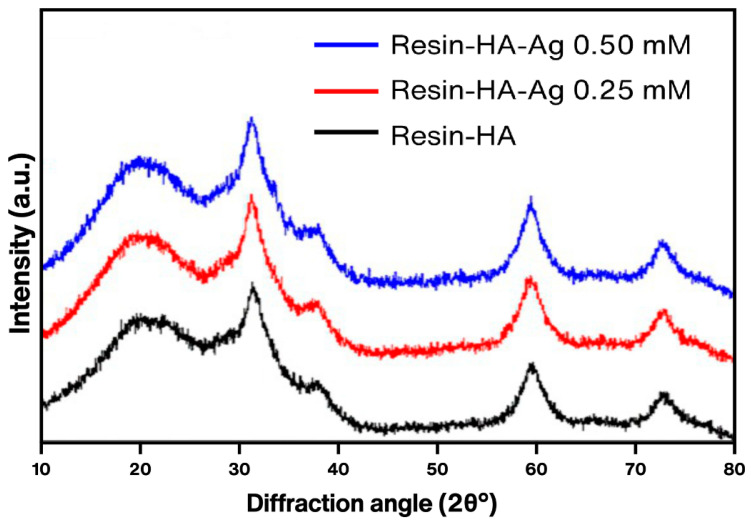
XRD patterns of the composite resin with hydroxyapatite and silver 0.50 mM (blue), silver 0.25 mM (red), and 0.00 mM (black).

**Figure 7 polymers-16-02017-f007:**
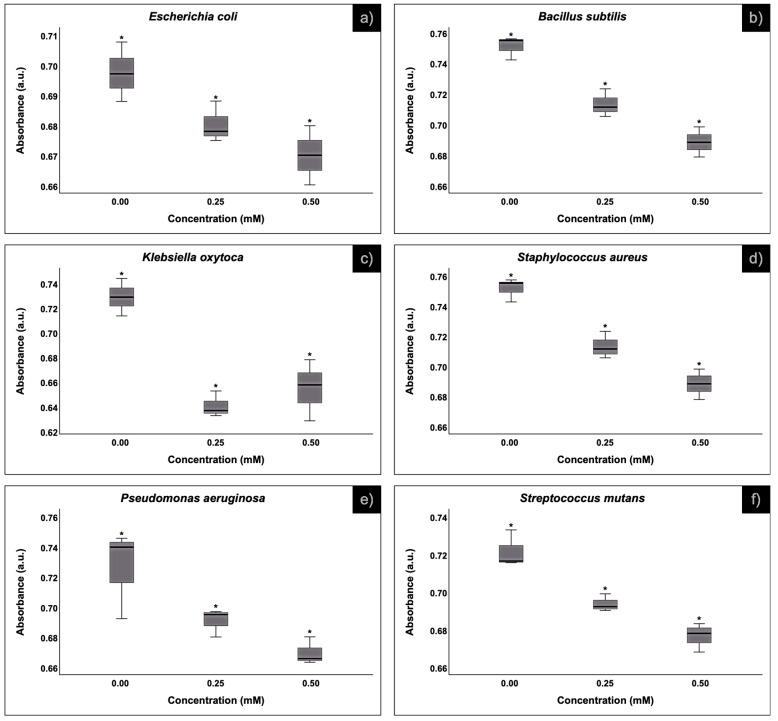
Box-plot of the inhibition behavior of the HA-AgNPs composite resin over 3 species of Gram-negative and 3 species of Gram-positive bacteria at 8 h. (**a**) *Escherichia coli*, (**b**) *Bacillus subtilis*, (**c**) *Klebsiella oxytoca*, (**d**) *Staphylococcus aureus*, (**e**) *Pseudomonas aeruginosa* and (**f**) *Streptococcus mutans*. (*) The asterisk on the box-plots stands for “significantly different”.

**Table 1 polymers-16-02017-t001:** Diffusion test of hydroxyapatite–silver composite resin.

Bacteria	Silver Nanoparticles Concentration		
	Resin with HA	0.25 mM	0.50 mM
	**Inhibition diameter (mm)**		
*Bacillus subtilis*	—±0.00	—±0.00	—±0.00
*Escherichia coli*	—±0.00	—±0.00	—±0.00
*Klebsiella oxytoca*	—±0.00	—±0.00	5.73 ± 0.50
*Pseudomonas aeruginosa*	—±0.00	—±0.00	—±0.00
*Staphylococcus aureus*	—±0.00	—±0.00	12.18 ± 0.97
*Streptococcus mutans*	—±0.00	5.35 ± 0.17	12.18 ± 1.32

**Table 2 polymers-16-02017-t002:** Literature review of research of the effect of AgNPs and AgNPs-HA composites on bacteria.

Material	Inoculum (CFU/mL)	Particle Size (nm)	Concentration AgNPs	Bacterium/(MIC)	Reference
HA-AgNPs	1.3 × 10^6^	2.65 ± 0.5	0.5 mM (0.027 mg/g), 0.25 mM (0.0135 mg/g)	*E. coli*/>0.5 mM *S. mutans*/>0.5 mM *E. aureus*/>0.5 mM *K. oxytoca*/>0 5 mM *P. aeruginosa*/>0.5 mM *B. subtilis*/>0.5 mM	Present work
Silver nanoparticles	1.0 × 10^4^	16	0, 20, 40, 60, 80 and 100 µg/mL	*E: coli*/20 µg/mL Inhibit colony-forming unit at 60 µg/mL	[49]
Bovine femur bone hydroxyapatite–silver nanoparticles	1.5 × 10^6–8^	8–20	0, 1, 3, and 5%	*E. coli*/3% *MRSA*/1%	[50]
Nano hydroxyapatite doped with silver	1.0 × 10^6^	12.0 ± 5.0	2, 2.5, 3, 3.6, 4.5 and 5.4%	*E. coli*/2% of 10 mm of inhibition *S. aureus*/2% 11 mm of inhibition	[51]
PCL-AgNPs	1.3 × 10^6^	5.6 ± 2.9	12.5 mM (0.012 mg/g), 25.0 mM (0.024 mg/g), 50.0 mM (0.046 mg/g) 100.0 mM (0.097 mg/g)	*E. coli*/>12.5 mM *E. aureus*/>12.5 mM *K. oxytoca.*/>12.5 mM *P. aeruginosa*/>12.5 mM	[52]
Silver doped hydroxyapatite	1.0 × 10^5^	-	0, 0.1, 0.3, 0.5, 0.7 M of AgNO_3_	*S. aureus*/0.1 M	[53]
HA-AgNPs	1.0 × 10^5^	-	0.01 M, 0.05 M, and 0.1 M of AgNO_3_	*E. coli*/0.01 M *S. aureus*/0.01 M	[54]
HA, AgNPs, Cotton	0.1 × 10^5^–10^6^	14	2.5 and 5%	*S. aureus*/*99%* of inhibition *E. coli*/*96%* of inhibition *C. albicans*/-*A. niger*/-	[55]

## Data Availability

The original contributions presented in the study are included in the article, further inquiries can be directed to the corresponding author.

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
