# Peer review of "Polymer-Based Hydroxyapatite–Silver Composite Resin with Enhanced Antibacterial Activity for Dental Applications"

_polymers, 2024, doi:10.3390/polym16142017_

Round 1
Reviewer 1 Report
Comments and Suggestions for Authors
Dear editor and the authors,
The paper describes hydroxyapatite-silver nanoparticle development and its application on bacterial species that cause problems for teeth. Even though there are similar papers in the literature (e.g. J. Wang et al. / Vacuum 152 (2018) 132e137; BioNanoScience (2023) 13:2215–2224 etc.), the surface chemistry of the silver nanoparticles are different, which can bring a difference on the performance of the composite structure. The experimental section was reviewed under the light of the cited articles. Therefore, the paper should meet with the readers. There are certain points need to be addressed before it gets published.
1- In the abstract, the second sentence must be the first sentence.
2- The first sentence can be removed, but in any case it should not be the first sentence
3- In Results section, citation required for the raman spectrum explanations.
4- The paper uses the term "silver resin", but Fiure 2d reveals silver nanoparticle. Can it be explained how the nanoparticles were named as silver resin?
5- Utilization of gallic acid must be justified.
Author Response
The paper describes hydroxyapatite-silver nanoparticle development and its application on bacterial species that cause problems for teeth. Even though there are similar papers in the literature (e.g. J. Wang et al. / Vacuum 152 (2018) 132e137; BioNanoScience (2023) 13:2215–2224 etc.), the surface chemistry of the silver nanoparticles are different, which can bring a difference on the performance of the composite structure. The experimental section was reviewed under the light of the cited articles. Therefore, the paper should meet with the readers. There are certain points need to be addressed before it gets published.
1- In the abstract, the second sentence must be the first sentence.
Response: Changes were made
2- The first sentence can be removed, but in any case it should not be the first sentence
Response: Changes was made
3- In Results section, citation required for the Raman spectrum explanations.
Response: Changes were made, reference is added
4- The paper uses the term "silver resin", but Figure 2d reveals silver nanoparticle. Can it be explained how the nanoparticles were named as silver resin?
Response: Figure 2 only corresponds to synthesis of silver nanoparticles
5- Utilization of gallic acid must be justified.
Response: Changes were made, justification is added
Reviewer 2 Report
Comments and Suggestions for Authors
The study addresses "Polymer-Based Hydroxyapatite-Silver Resin Composite with Enhanced Antibacterial Activity for Dental applications".
The study was generally well designed. However, the following points need to be clarified.
The effect and importance of the species that cause infections should be explained in the introduction . Also, the sentence "Three of the bacteria are ........... immunosuppressed patients [30]" (lines 139-142) should be moved to the introduction section.
The source, code or number of the strains used should be given. And even information is needed as to which collection it was obtained from. It would be good if a reference was also given.
The statistical analysis should be given in a separate paragraph.
The references given in Table 2 should be mentioned in the introduction. Table 2 should be removed.
Lines 328-337: This section should be rewritten, taking into account other research results. Clear and unambiguous expressions should be used.
Author Response
The study addresses "Polymer-Based Hydroxyapatite-Silver Resin Composite with Enhanced Antibacterial Activity for Dental applications".
The study was generally well designed. However, the following points need to be clarified.
The effect and importance of the species that cause infections should be explained in the introduction . Also, the sentence "Three of the bacteria are ........... immunosuppressed patients [30]" (lines 139-142) should be moved to the introduction section.
Response: Changes were made, new information was added in introduction section
The source, code or number of the strains used should be given. And even information is needed as to which collection it was obtained from. It would be good if a reference was also given.
Response: Changes were made, new information was added.
The statistical analysis should be given in a separate paragraph.
The references given in Table 2 should be mentioned in the introduction. Table 2 should be removed.
Response: We believe the Table should be in the discussion section, as it helps to compare our results with results from other investigations.
Lines 328-337: This section should be rewritten, taking into account other research results. Clear and unambiguous expressions should be used.
Response: Changes were made, the text was rewritten.
Reviewer 3 Report
Comments and Suggestions for Authors
The article "Polymer-Based Hydroxyapatite-Silver Resin Composite with Enhanced Antibacterial Activity for Dental Applications." devoted to development of new antibacterial resin. The obtained results show highly competitive properties compared to existing works, which indicates the high novelty and significant contribution of this work. However, there are a number of shortcomings in the work, which should be corrected to improve the work: 1. Materials and methods are written in too much detail in the abstract. 2. What does the text in the abstract "Chemical analysis revealed the presence of silicon and monoclinic zirconia in the composite resin" mean for your work? Why do the authors present this result separately? Shouldn't it be described in detail in the abstract? 3. PVP in the introduction needs to be deciphered. 4. There is no materials item in materials and methods. In this paragraph it is also necessary to specify the manufacturer of the components used, indicating the city and country of the manufacturer. 5. In the article the text is sometimes in yellow color, needs to be corrected. 6. Figure 1E. Elements on the insert are difficult to read. 7. Figure 2b. The distribution between 1 and 100 nm is poorly visible. Currently, the graph shows that the average size is about 30-50 nm, not 2.65. I recommend increasing the range from 1 to 100 nmnm if possible. 8.Lines 257-261. It says that a concentration of 0.25 and 0.5 mM is used. But it does not say the concentration of what substance. 9.Figure 7 is in a different chapter of the paper. I recommend moving figure 7 closer to the text that describes these figures. 10. Figure 7. Absence of absorbance for bacteria without composite resin. This does not allow us to conclude about the influence of composite resin.
Author Response
The article "Polymer-Based Hydroxyapatite-Silver Resin Composite with Enhanced Antibacterial Activity for Dental Applications." devoted to development of new antibacterial resin. The obtained results show highly competitive properties compared to existing works, which indicates the high novelty and significant contribution of this work. However, there are a number of shortcomings in the work, which should be corrected to improve the work:
- Materials and methods are written in too much detail in the abstract.
The abstract has been slightly rewritten to accommodate less details.
- What does the text in the abstract "Chemical analysis revealed the presence of silicon and monoclinic zirconia in the composite resin" mean for your work? Why do the authors present this result separately? Shouldn't it be described in detail in the abstract?
Data of XRD showed the presence of silica and zirconia in the resin, and this information was included for the composition of the resin. Most of the dental resins do have these components, however, the information was removed from the abstract as it is not key information and appears only in results.
- PVP in the introduction needs to be deciphered.
The PVP acronym has been changed for the name of the compound, polyvynilpirrolidone.
- There is no materials item in materials and methods. In this paragraph it is also necessary to specify the manufacturer of the components used, indicating the city and country of the manufacturer.
Information missing in the description of the compounds used has been added. The structure of the section has been maintained, were materials are written within the prose togheter with the methods and devices used on the synthesis.
- In the article the text is sometimes in yellow color, needs to be corrected.
The text in yellow was changed to not have the yellow background.
- Figure 1E. Elements on the insert are difficult to read.
The Figure 1 has been modify to increase the size of the EDX analysis and its text.
- Figure 2b. The distribution between 1 and 100 nm is poorly visible. Currently, the graph shows that the average size is about 30-50 nm, not 2.65. I recommend increasing the range from 1 to 100 nmnm if possible.
A correction was made on the size of the particles to be 26.5 nm instead of 2.65 nm which is incorrect.
8.Lines 257-261. It says that a concentration of 0.25 and 0.5 mM is used. But it does not say the concentration of what substance.
It refers to the concentration of the silver. The text has been changed to include this information.
9.Figure 7 is in a different chapter of the paper. I recommend moving figure 7 closer to the text that describes these figures.
The figure was moved to be next to the paragraph were it is mentioned.
- Figure 7. Absence of absorbance for bacteria without composite resin. This does not allow us to conclude about the influence of composite resin.
This part was not changed, as we used the resin alone as a control, since there is no precedent on the antibacterial effect on the resin.

Round 2
Reviewer 2 Report
Comments and Suggestions for Authors
The article can be accepted.